# Process Corresponding Implications Associated with a Conclusive Model-Fit Current-Voltage Characteristic Curves

**Hsin-Chia Yang * and Sung-Ching Chi**

Department of Electronic Engineering, Ming Hsin University of Science and Technology, Hsinchu 30401, Taiwan; chisc@must.edu.tw
* Correspondence: hcyang@must.edu.tw

**Abstract:** NFinFET transistors with various fin widths (110 nm, 115 nm, and 120 nm) are put into measurements, and the data are collected. By using the modified model, the measure data is fitted. Several parameters in the formula of modified model are determined to make both the measured data and the fitting data almost as close as possible. Those parameters are listed and analyzed, including $k_N$ (proportional to channel width and gate oxide capacitor, and inversely proportional to the channel length) $\lambda$ (the inverse of Early Voltage), and sometimes $V_{th}$ (Threshold Voltage). By $k_N$, the appropriate process control can be high lighted, the corresponding channel concentration can be calculated and thus many implicit physical quantities may be exploited.

**Keywords:** FinFET; early voltage; channel length and width

## 1. Introduction

Source-Drain leakage current, known as one of the short channel effects, is successfully suppressed on FinFET transistors. The structure of FinFET looks like an emerging 3-dimensional "I" character with Source and Drain as the two ends and the channel in between. The applied bias on Gate poly-silicon crossing over the channel depletes the whole slim channel and builds up a barrier in between Source and Drain, which thus blocks or prevents the possibility of leakage current. The process is achieved because of the good conformality (step coverage) of chemical vapor deposition by flowing $SiH_4$ (silane) at chosen flow rate under certain pressure at an appropriate temperature in kinetic regime for a good deposition rate [1–5].

Furthermore, the electrical performance of FinFET transistors may be also enhanced either by high dielectric constant gate oxide or by high mobility channel. On one hand, the capacitance of the gate capacitor is to be raised if the dielectric of the gate oxide, mainly silicon dioxide, can be replaced with Hf-mixed tantalum oxide whose dielectric constant is about 5 times of that of N-mixed silicon dioxide. For instance, 90 nm-process devices can be equivalently reduced to 14 nm-process devices, which is quite encouraging once the process is production-worthy. On the other hand, the mobility of silicon channel may be promoted by even 2.5 to 4 times as SiGe is technically and sophisticatedly introduced stack by stack. The above advanced techniques and other options are definitely promising and achievable, making FinFET continuously popular as currently [6–12].

However, the electrical performances, mainly manifested in current-versus-voltage characteristic curves (I–V curves), are thus put to be parameter-extracted in the model, which takes advantage of sophisticated equivalent circuits for academic and industrial uses. Nevertheless, the measured I–V curves are speculated to be also fitted by the "modified" conventional formula [13–18].

In this study, the "modified" conventional I–V characteristic curve formula naively generates fitting data to fit as measured ($I_{DS}$, $V_{DS}$) and ($I_{DS}$, $V_{GS}$) data by choosing appropriate parameter values, such as the threshold voltage ($V_{th}$), lambda ($\lambda$) which is inversely

proportional to the absolute value of Early Voltage ($V_A$), and $k_N$ which is proportional to the total width of the channel and inversely proportional to the channel length.

## 2. Preparation and Measurements

### 2.1. Preparation

The epi-silicon layer is deliberately ionic dry etched with various 3-dimensional sizes, such as 110 nm, 115 nm, and 120 nm wide and corresponding 9 times high fin channel. On both ends of the channel are Source and Drain, looking like a letter "I". The epi-silicon fin channel is grown with ultra thin gate oxide and covered with arsenic heavily doped poly-silicon.

### 2.2. Fitting $I_{DS}$-$V_{DS}$ and $I_{DS}$-$V_{GS}$

The two-regime modified conventional formulas for FinFET transistors are expressed in the following:

$$I_{DS}(\text{Triode}) = k_N[(V_{GS} - V_{th})V_{DS} - V_{DS}{}^2/2](1 + \lambda V_{DS}) \qquad (1)$$

$$I_{DS}(\text{Saturation}) = \frac{k_N}{2}(V_{GS} - V_{th})^2(1 + \lambda V_{DS}) \qquad (2)$$

where

$$k_N = \frac{C_{ox}W_{eff}\mu}{L_o}, \ \lambda = \frac{1}{|V_A|}, \ \text{and } \alpha(\text{gate leakage coefficient}).$$

where $V_A$ is Early Voltage, $C_{ox}$ is gate oxide capacitance, and $W_{eff}$ = 19$W_o$.

Equation (1) works as $V_{DS}$ is less than ($V_{GS} - V_{th}$) while Equation (2) works as $V_{DS}$ is larger than ($V_{GS} - V_{th}$). And those parameters are predominantly determined to minimize the deviation (delta, $\delta$) as follows in Equation (3):

$$\delta = \sqrt{\frac{\sum\limits_{i=1}^{N}(I_{fitting} - I_{measured})_i{}^2}{N}} \qquad (3)$$

## 3. Results and Discussion

The minimum deviation (delta, $\delta$ in Equation (3) is used to determine the chosen parameters. For example, the measured characteristic curve on the device denoted by W120L240 (fin width = 120 nm, channel length = 240 nm) at $V_G$ = 1.0 V may be well fitted by choosing the three parameters $V_{th}$ = 0.101 V, $k_N$ = 1.09 × 10$^{-4}$ A/V$^2$, and lambda = 0.139 in Equations (1) and (2) as shown in Figure 1a–c, where the minimum deviation value is found to be 3.79569 × 10$^{-7}$ A.

Therefore by using the same skill, all the parameters in Equations (1) and (2) are determined to fit the measured characteristic curves except the ones in Figure 2a,b. where the two transistors, denoted by W110L100 (Fin width = 110 nm, Channel length = 100 nm) and W110L120 (Fin width = 110 nm, Channel length = 120 nm), do not look like FET and are not worth fitting. All the other fitting results are shown in Figures 2c, 3a–c and 4a–c. The final minimum delta's at various $V_G$'s. range from 3.25 × 10$^{-8}$ Ampere to 1.36 × 10$^{-6}$ Ampere, which are engineering acceptable.

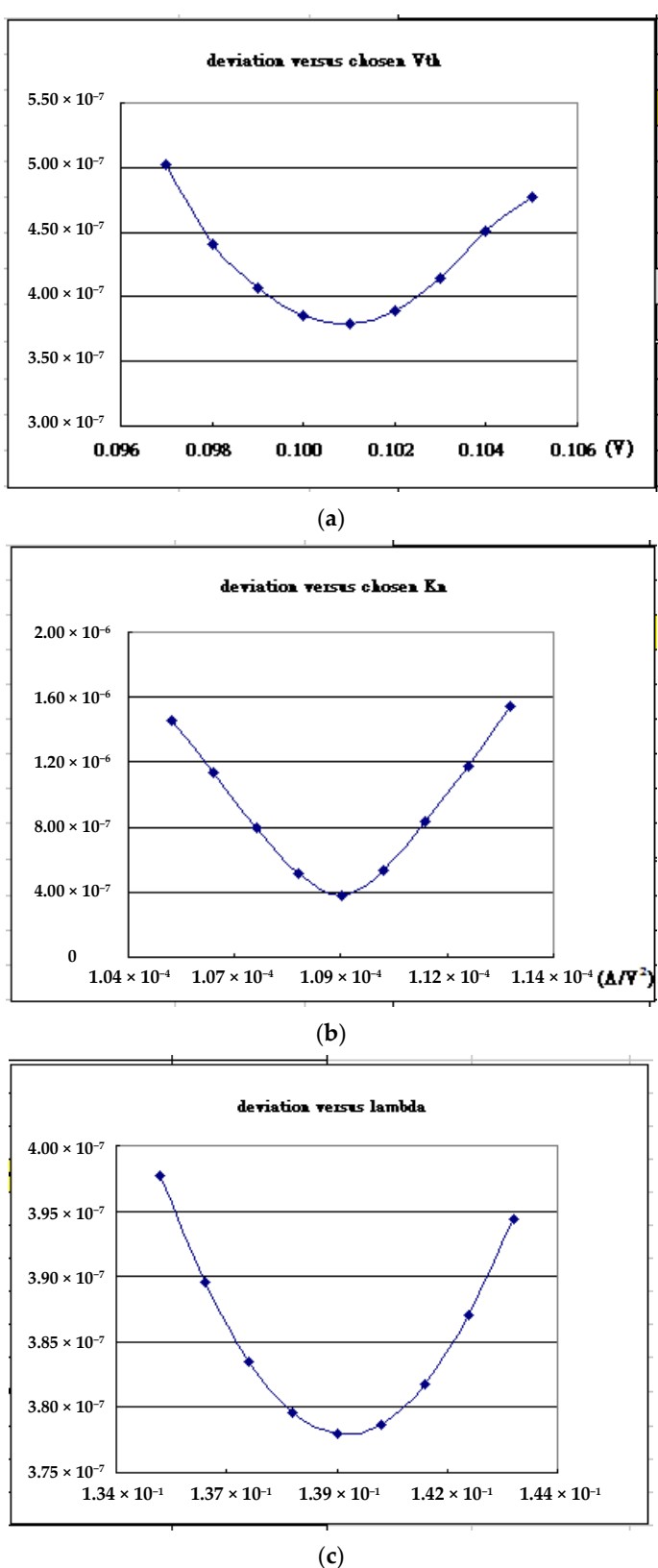

(a)

(b)

(c)

**Figure 1.** Three parameters ($V_{th}$, $K_n$, lambda) at $V_G$ = 1.0 V on W120L240 (fin width = 120 nm, and channel length = 240 nm) are determined through minimum deviation technique with the minimum deviation = 3.79569 × 10$^{-7}$ A at (**a**) $V_{th}$ = 0.101 V as in deviation versus $V_{th}$, (**b**) $k_N$ = 1.09 × 10$^{-4}$ A/V$^2$ as in deviation verses $k_N$, and (**c**) lambda = 0.139 as in deviation versus lambda.

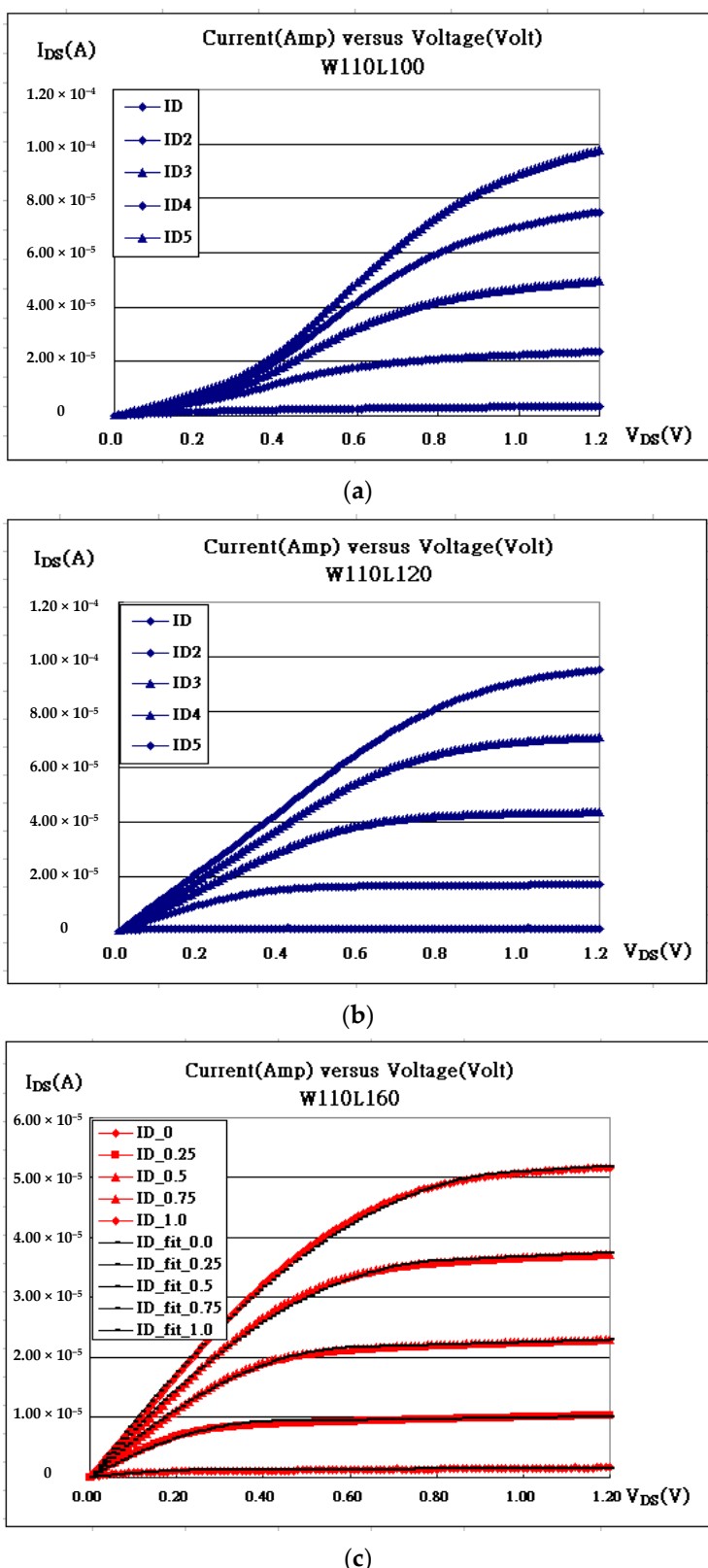

**Figure 2.** $I_{DS}$-$V_{DS}$ characteristic curves and the corresponding fitting, including (**a**) W110L100, (**b**) W110L120, (**c**) W110L160.

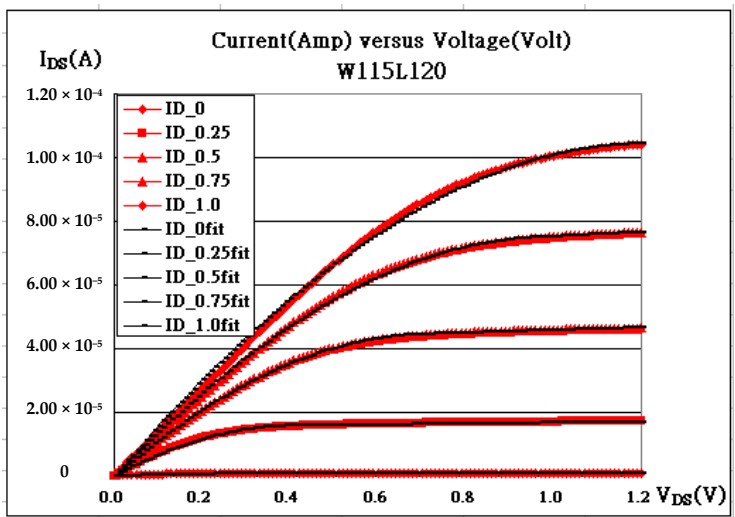

(**a**) with delta ranging from $1.01 \times 10^{-7}$ to $1.36 \times 10^{-6}$ amperes

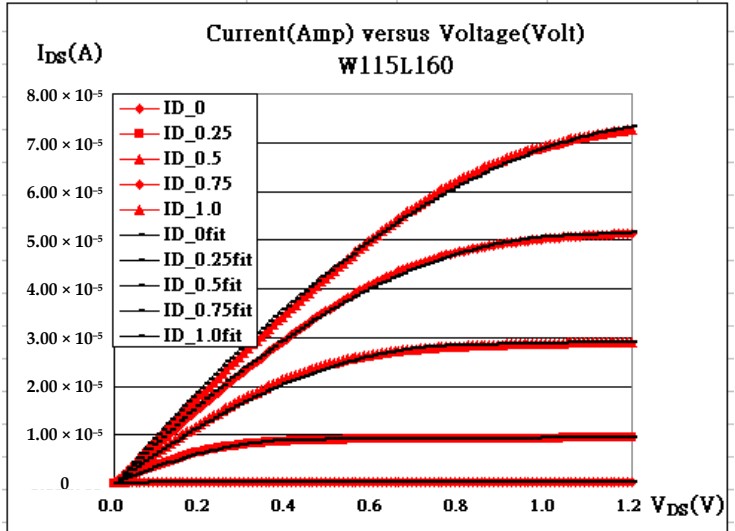

(**b**) with delta ranging from $2.75 \times 10^{-8}$ to $9.21 \times 10^{-7}$ amperes

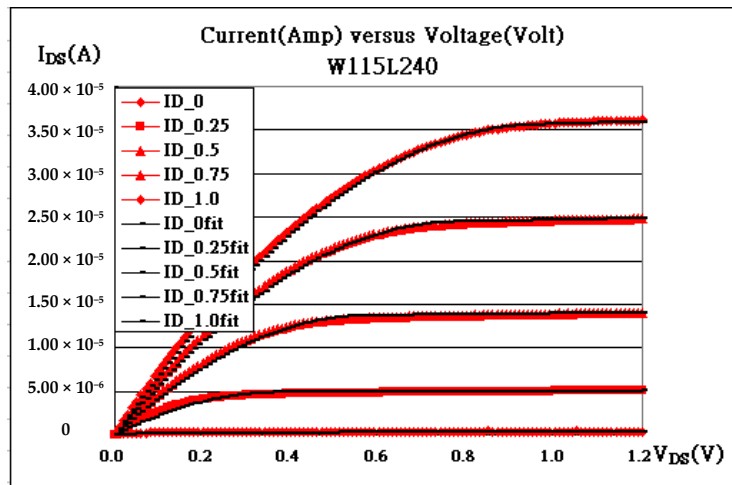

(**c**) with delta ranging from $1.07 \times 10^{-7}$ to $3.37 \times 10^{-7}$ Amperes

**Figure 3.** $I_{DS}$-$V_{DS}$ characteristic curves and the corresponding fitting, including (**a**) W115L120, (**b**) W115L160, (**c**) W115L240.

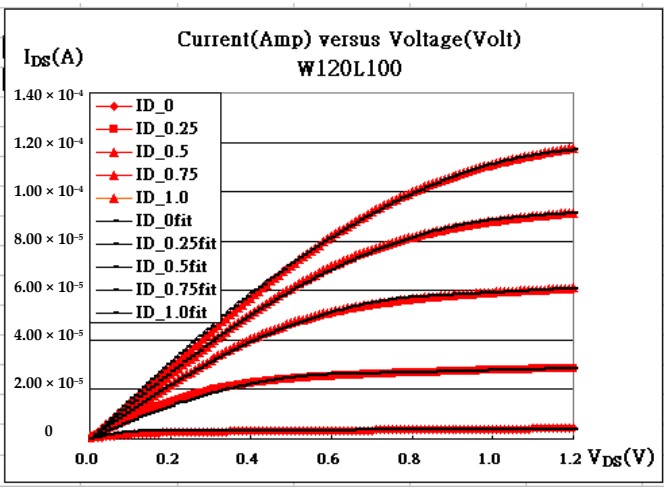

(**a**) with delta ranging from 3.72 × 10⁻⁷ to 1.20 × 10⁻⁶ amperes

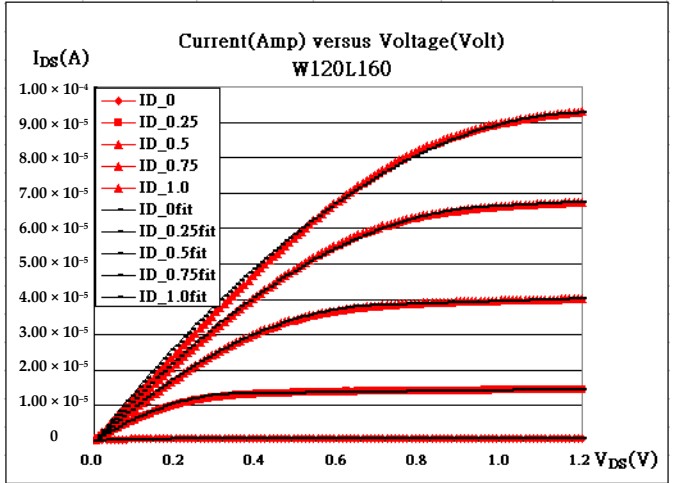

(**b**) with delta ranging from 1.03 × 10⁻⁷ to 1.22 × 10⁻⁶ amperes

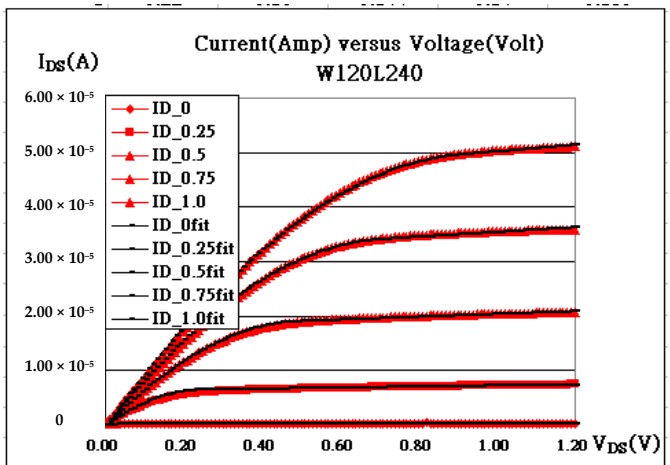

(**c**) with delta ranging from 3.25 × 10⁻⁸ to 3.77 × 10⁻⁷ amperes

**Figure 4.** $I_{DS}$-$V_{DS}$ characteristic curves and the corresponding fitting, including (**a**) W120L110, (**b**) W120L160, and (**c**) W120L240.

In addition, all determined $k_N$'s may be specifically listed as in Tables 1–3 at different sizes and at different $V_G$'s [12]. In the modified conventional formula in Equations (1) and (2), $k_N$ is supposed to be inversely proportional to the channel length. And $k_N$ is also proportional to the channel width ($W_{channel}$) and, thus the width of the fin ($W_{fin}$), because

$W_{channel} = (9 + 9 + 1) \, W_{fin} = 19 W_{fin}$. If $k_N$ is plotted against $W_{fin}$ or $1/L_{channel}$, straight lines are then expected. Unfortunately, most of them fail the testing except on the devices with fin width = 120 nm at $V_G = 1.0$ V partly as found in Figure 5. Of course, at $V_G = 0.0$ V, the transistors, which are still on and may have pretty low threshold voltage or even negative one, are not discussed in this paper. So the applied voltages to Gates depleting the fin channels and making the devices less leaky draw much attention. In Figure 4a, the straight line proves the feasibility at $V_G = 1.0$ V with fin width = 120 nm, while the other two graphs, Figure 4b,c do not. The non-straight lines might be due to either over-depletion or under-depletion. Over-depletion means that the depletion regions from both sides of the fin overlap resulting to disturbing the blocking function. On the other hand, the depletion regions do not completely cover all the fin for under-depletion and the leakage current gets apparent.

**Table 1.** $k_N$ values with Fin Width = 120 nm.

| Gate Bias | W120L240_fit | w120L160_fit | W120L100_fit |
|---|---|---|---|
| $V_G = 1.00$ V | $1.09000 \times 10^{-4}$ | $1.25000 \times 10^{-4}$ | $1.43000 \times 10^{-4}$ |
| $V_G = 0.75$ V | $1.27000 \times 10^{-4}$ | $1.30000 \times 10^{-4}$ | $1.40000 \times 10^{-4}$ |
| $V_G = 0.50$ V | $1.48000 \times 10^{-4}$ | $1.48000 \times 10^{-4}$ | $1.46000 \times 10^{-4}$ |
| $V_G = 0.25$ V | $2.10000 \times 10^{-4}$ | $2.10000 \times 10^{-4}$ | $1.27000 \times 10^{-4}$ |
| $V_G = 0.00$ V | $8.70000 \times 10^{-4}$ | $8.80000 \times 10^{-5}$ | $3.10000 \times 10^{-4}$ |

**Table 2.** $k_N$ values with Fin Width = 115 nm.

| Gate Bias | W115L240_fit | W115L160_fit | W115L120_fit |
|---|---|---|---|
| $V_G = 1.00$ V | $7.60000 \times 10^{-5}$ | $8.14000 \times 10^{-5}$ | $1.40000 \times 10^{-4}$ |
| $V_G = 0.75$ V | $8.00000 \times 10^{-5}$ | $8.65000 \times 10^{-5}$ | $1.50000 \times 10^{-4}$ |
| $V_G = 0.50$ V | $8.18000 \times 10^{-5}$ | $8.40000 \times 10^{-5}$ | $1.70000 \times 10^{-4}$ |
| $V_G = 0.25$ V | $7.00000 \times 10^{-5}$ | $9.00000 \times 10^{-5}$ | $1.90000 \times 10^{-4}$ |
| $V_G = 0.00$ V | $1.00000 \times 10^{-5}$ | $1.00000 \times 10^{-5}$ | $1.00000 \times 10^{-5}$ |

**Table 3.** $k_N$ values with Fin Length = 160 nm.

| Gate Bias | w120L160_fit | W115L160_fit | W110L160_fit |
|---|---|---|---|
| $V_G = 1.00$ V | $1.25000 \times 10^{-4}$ | $8.14000 \times 10^{-5}$ | $1.00000 \times 10^{-4}$ |
| $V_G = 0.75$ V | $1.30000 \times 10^{-4}$ | $8.65000 \times 10^{-5}$ | $1.03000 \times 10^{-4}$ |
| $V_G = 0.50$ V | $1.48000 \times 10^{-4}$ | $8.40000 \times 10^{-5}$ | $1.10000 \times 10^{-4}$ |
| $V_G = 0.25$ V | $2.10000 \times 10^{-4}$ | $9.00000 \times 10^{-5}$ | $1.00000 \times 10^{-4}$ |
| $V_G = 0.00$ V | $8.80000 \times 10^{-5}$ | $1.00000 \times 10^{-5}$ | $2.50000 \times 10^{-5}$ |

The three transistors with 120 nm fin width at $V_G = 1.0$ V provide valuable information, i.e., the whole fin may be totally depleted without interference from the other side of the applied bias. One or the other side of the applied Gate bias depleted 60 nm, which is a half of the fin width, as shown in Figure 6, where the energy band appears with 1.12 eV energy gap. The so-called p-type channel or substrate is interpreted as boron dopants-in the silicon lowering the Fermi energy. Once the applied Gate bias bents the intrinsic Fermi energy down below the Fermi energy, the region becomes strongly inversed to n-type semi-conductor making the channel conductive. The thickness of the conductive layer may as well be calculated by solving Maxwell's Equations as stated in Equations (4) and (5) with W set to 60 nm. The channel concentration is then estimated to be $p = 3.66 \times 10^{23}$ (m$^{-3}$), which is substituted into $(K_B T) \ln(p/n_i)$ to obtain 0.438 eV (energy difference between $E_{F(i)}$ and $E_F$). Therefore, the strong inversion layer is determined to be 203 angstroms, which was one half of the parabolic curve (in Equation (4)) in Figure 6, and is surprisingly about one third of the whole fin width.

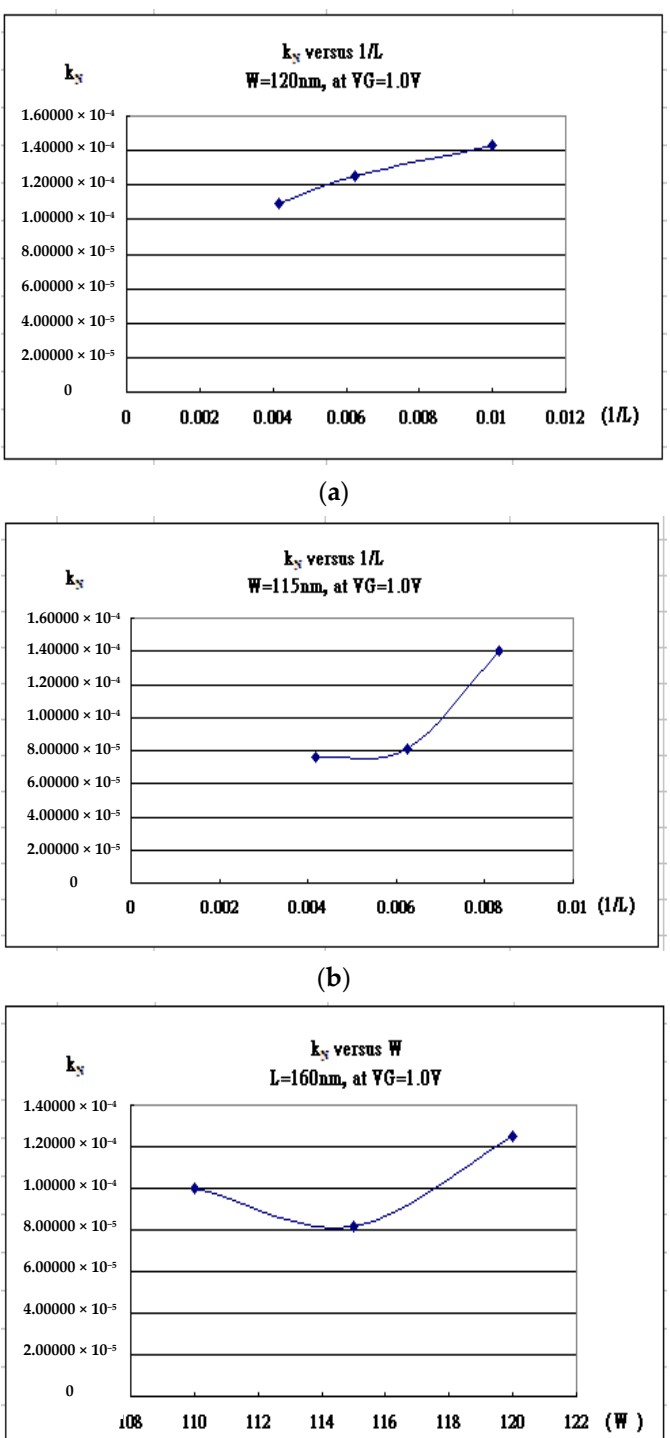

**Figure 5.** $k_N$ is supposed to be proportional to the channel width or the inverse of channel length. (**a**) $k_N$ versus 1/L with fin width = 120 nm, (**b**) $k_N$ versus 1/L with fin width = 115 nm, and (**c**) $k_N$ versus W with channel length = 160 nm.

$$\nabla \cdot \vec{E} = \frac{\rho}{\varepsilon} \Rightarrow \frac{dE}{dx} = -\frac{ep}{\varepsilon}$$
$$\Rightarrow E_p = -\frac{ep}{\varepsilon}(x - W)$$

$$\Rightarrow V_G = \frac{1}{2}\frac{ep}{\varepsilon}(W)^2 \tag{4}$$

$$\Rightarrow p = \frac{2\varepsilon V_G}{eW_{1/2}{}^2} \tag{5}$$

where

$$K_B = 1.38 \times 10^{23} \text{ J/K},$$
$$e = 1.69 \times 10^{-19} \text{ Coul},$$
$$n_i = 1.45 \times 10^{16}/\text{m}^3,$$
$$T = 298 \text{ K},$$
$$\varepsilon = 11.9 \times 8.85 \times 10^{-12} \text{ F/m}$$

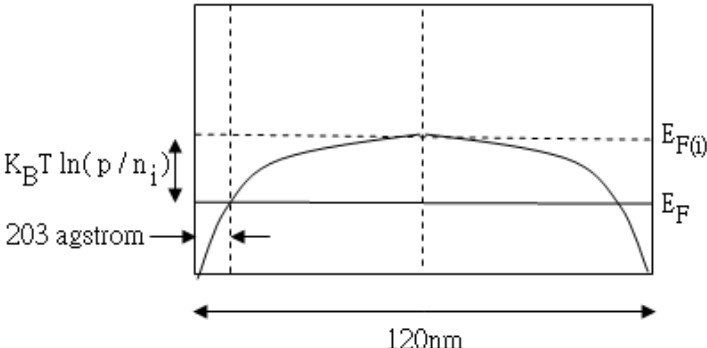

**Figure 6.** Silicon energy gap addressing bent intrinsic Fermi Energy, p-type Fermi Energy, and the strong-inversed layer from p-type to n-type.

## 4. Conclusions

The modified conventional current-voltage formula has demonstrated the fitting capability of the electrical characteristic curves. Once the parameters are determined through engineering fitting, those parameters are advisable to understand the implicit physics underlying the FET transistor. The current-voltage characteristic fitting simply obeys the modified formula in Equations (1) and (2) and bases on the minimum deviation without first considering the underlying physics. After all crucial parameters, e.g., $k_N$, $V_{th}$, and lambda, are determined, the analyses were available. In this paper, $k_N$ in Equations (1) and (2) was deliberately examined first, and the thickness of the layer associated with strong inversion is then successfully solved. At the same time, $V_{th}$ and lambda offering some more information about depletion region and leakage current may be actually expected.

Nevertheless, the model, conventionally established much earlier many decades ago, has been using equivalent circuits to approach to the measured data, and enjoy many fruitful achievements [17]. But tedious work and convergence still have to be taken into account.

In a word, fitting skill proposes another possibility to analyze the transistor. Many approaches, such as lambda corresponding to leakage current, the common threshold voltage requirement, and the common $k_N$ requirement, are to be associated with one another. Those analyses are quite intriguing and will be discussed in the near future.

**Author Contributions:** Conceptualization, H.-C.Y. and S.-C.C.; methodology, H.-C.Y. and S.-C.C.; software, H.-C.Y. and S.-C.C.; validation, H.-C.Y. and S.-C.C.; formal analysis, H.-C.Y. and S.-C.C.; investigation, H.-C.Y. and S.-C.C.; resources, H.-C.Y. and S.-C.C.; data curation, H.-C.Y. and S.-C.C. All authors have read and agreed to the published version of the manuscript.

**Funding:** This research received no external funding.

**Institutional Review Board Statement:** Not available.

**Informed Consent Statement:** Not available.

**Data Availability Statement:** Not available.

**Conflicts of Interest:** The authors declare no conflict of interest.

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
