# Peer review of "Process Corresponding Implications Associated with a Conclusive Model-Fit Current-Voltage Characteristic Curves"

_applsci, doi:10.3390/app12010462_

Round 1

Reviewer 1 Report

Dear Authors,

Introduction part should be developed with current novel publications. The number of references is very few, it would be useful to increase it. The conclusion part should be written in more detail.

Author Response

  1. All the texts in the paper have been reviewed once more, and some words in red are newly input.
  2. The format in Reference has been corrected and the cited papers in red were newly added. 

Reviewer 2 Report

The manuscript presents research related to the modeling of the electrical characteristics of NFinFET transistors. Based on the measurements, model parameters are determined to be used for the description of the main characteristics and hence to enter into the explanation of the physical processes in the semiconductor structure. In order to acquire a type of manuscript acceptable for publication, it needs serious revision and editing. My main remarks are the following:

  • it would be good to make an in-depth literature study of the state of the problem and on this basis to indicate the unresolved problems and to outline the directions of the research;
  • there is essentially no analysis of the results. No summaries, conclusions and recommendations have been made. In the end, there is a conclusion that very simply and quickly says what has been done;
  • there is no comparison with other methods and manuscripts that deal with similar topics. Thus, it is not clear what the advantages and contributions of the authors are;
  • it is necessary to review the formatting of the entire text;
  • no verification of the results.

Author Response

  1. The fitting using several parameters, which address various specific physical meanings, is used to identify each parameter's contribution.
  2. It is impressive and worthy to obtained the concentration of the channel, which is used to estimate the thickness of the strongly inverted layer.

Round 2

Reviewer 2 Report

Unfortunately, the authors have taken my remarks and recommendations lightly. There is a lack of in-depth literature review to identify unresolved issues and highlight the authors' contributions. The corrections made are very superficial and do not change the manuscript in essence. On the other hand, the research task has a good potential for development and makes it possible to obtain a quality manuscript. In my opinion, the use of only 8 sources for references in an article in an authoritative journal is unacceptable.

Author Response

Thank you for your encouragement. 

Sorry to make you down when the replied letter seemed to be superficial.

But if possible, could you give us some reference of in-depth literature so that we could follow and make our best for the next improvement.

Round 3

Reviewer 2 Report

Hello, I hope you found my comments and recommendations useful. Basically, when submitting an article in a reputable journal, in addition to the results (which are interesting and indisputable to you) it is very important to make an in-depth literature review to find out what are the unsolved problems and what is your research goal and task. In this regard, I am sending you several manuscripts on the subject that could be useful to you and I think they are written in a very good style.

  1. Lu, P.; Yang, C.; Li, Y.; Li, B.; Han, Z. Three-Dimensional TID Hardening Design for 14 nm Node SOI FinFETs. Eng 20212, 620-631. https://doi.org/10.3390/eng2040039
  2. Huang, P.; Ma, C.; Wu, Z. Fast Dynamic IR-Drop Prediction Using Machine Learning in Bulk FinFET Technologies. Symmetry 202113, 1807. https://doi.org/10.3390/sym13101807
  3. Li, Y.; Zhao, F.; Cheng, X.; Liu, H.; Zan, Y.; Li, J.; Zhang, Q.; Wu, Z.; Luo, J.; Wang, W. Four-Period Vertically Stacked SiGe/Si Channel FinFET Fabrication and Its Electrical Characteristics. Nanomaterials 202111, 1689. https://doi.org/10.3390/nano11071689
  4. Li, Y.; Zhao, F.; Cheng, X.; Liu, H.; Zan, Y.; Li, J.; Zhang, Q.; Wu, Z.; Luo, J.; Wang, W. Four-Period Vertically Stacked SiGe/Si Channel FinFET Fabrication and Its Electrical Characteristics. Nanomaterials 202111, 1689. https://doi.org/10.3390/nano11071689
  5. Wang, M.-C.; Hsieh, W.-C.; Lin, C.-R.; Chu, W.-L.; Liao, W.-S.; Lan, W.-H. High-Drain Field Impacting Channel-Length Modulation Effect for Nano-Node N-Channel FinFETs. Crystals 202111, 262. https://doi.org/10.3390/cryst11030262
  6. Lee, J.; Park, T.; Ahn, H.; Kwak, J.; Moon, T.; Shin, C. Prediction Model for Random Variation in FinFET Induced by Line-Edge-Roughness (LER). Electronics 202110, 455. https://doi.org/10.3390/electronics10040455

Author Response

The introduction, the conclusion, and the reference were modified a little. Thank you a lot for the advice.
